# Preclinical Testing of Radiopharmaceuticals for the Detection and Characterization of Osteomyelitis: Experiences from a Porcine Model

**DOI:** 10.3390/molecules26144221

**Published:** 2021-07-12

**Authors:** Aage Kristian Olsen Alstrup, Svend Borup Jensen, Ole Lerberg Nielsen, Lars Jødal, Pia Afzelius

**Affiliations:** 1Department of Nuclear Medicine & PET, Aarhus University Hospital, DK-8200 Aarhus, Denmark; 2Department of Clinical Medicine, Aarhus University, DK-8200 Aarhus, Denmark; 3Department of Nuclear Medicine, Aalborg University Hospital, DK-9000 Aalborg, Denmark; svbj@rn.dk (S.B.J.); lajo@rn.dk (L.J.); 4Department of Chemistry and Biosciences, Aalborg University, DK-9220 Aalborg, Denmark; 5Department of Veterinary and Animal Sciences, University of Copenhagen, DK-1165 Copenhagen, Denmark; olelerbergnielsen@gmail.com; 6Zealand Hospital, Køge, Copenhagen University Hospital, DK-4600 Køge, Denmark; pafz@regionsjaelland.dk

**Keywords:** animal osteomyelitis model, imaging, inoculation, PET, pig, preclinical, tracers

## Abstract

The development of new and better radioactive tracers capable of detecting and characterizing osteomyelitis is an ongoing process, mainly because available tracers lack selectivity towards osteomyelitis. An integrated part of developing new tracers is the performance of in vivo tests using appropriate animal models. The available animal models for osteomyelitis are also far from ideal. Therefore, developing improved animal osteomyelitis models is as important as developing new radioactive tracers. We recently published a review on radioactive tracers. In this review, we only present and discuss osteomyelitis models. Three ethical aspects (3R) are essential when exposing experimental animals to infections. Thus, we should perform experiments in vitro rather than in vivo (Replacement), use as few animals as possible (Reduction), and impose as little pain on the animal as possible (Refinement). The gain for humans should by far exceed the disadvantages for the individual experimental animal. To this end, the translational value of animal experiments is crucial. We therefore need a robust and well-characterized animal model to evaluate new osteomyelitis tracers to be sure that unpredicted variation in the animal model does not lead to a misinterpretation of the tracer behavior. In this review, we focus on how the development of radioactive tracers relies heavily on the selection of a reliable animal model, and we base the discussions on our own experience with a porcine model.

## 1. Introduction

Osteomyelitis is an infection of the bone and bone marrow. It is a common disease that is not always treated effectively with systemic antibiotics. Local surgical intervention is also often needed [1]. Scintigraphy and positron emission tomography (PET) offers non-invasive procedures that can localize osteomyelitis without any significant intervention or discomfort for the patient [1]. As discussed in our recent published review [2], the tracers that are in general use today, such as labelled leukocytes for single-photon emission computed tomography (SPECT; 3D scintigraphy) or the glucose analogue [^18^F]FDG for PET, are not optimal. The PET tracer [^18^F]FDG has a leading role in oncology [3] but is not specifically designed for infections and may also accumulate in relation to sterile inflammation. New tracers are needed for the specific localization of osteomyelitis, and they are also required to provide knowledge about infection characteristics, such as extent, activity, and stage. Therefore, research into the better imaging of infections is ongoing [1]. In a previously published paper in *Molecules*, we reviewed the present radiotracers used for bone marrow infection imaging [2]. Here, we review tests of tracers in osteomyelitis animal models based on our experience.

Knowledge of how well a potential tracer performs is essential. The clinical use of a tracer always involves prior animal testing. A wish to completely abandon animal testing is noble, but the physiology of a complex organism cannot in all aspects be limited to what goes on in a petri dish. However, the goal does not justify any means. The use of research animals should be guided by what is known as the three Rs formulated by Russell and Burch [4]:REPLACING animal testing by other methods when possible. For example, cell cultures can test if a theoretically promising tracer indeed binds to the expected cells. If that tracer does not even adhere to the cells as expected, there is no reason to proceed with animal experiments.REDUCING the number of animals involved. Planning the experiment should include a selection of relevant data and their analysis in a way that will make it possible to reject or prove the hypothesis and at the same time reduce the number of experimental animals to a minimum. However, it is also possible to use too few animals. If the scientific question cannot be answered or elucidated, the animals used are wasted.REFINEMENT of the animal model so that the involved discomfort or suffering is as limited as possible for those animals that are needed to be part of the experiments.

Using healthy animals allows an examination in a healthy physiological setting of the general kinetics, possible metabolism, and biodistribution of the tracer. In a disease model, animal testing provides an evaluation of the ability of the tracer to visualize disease, for example, osteomyelitis, in vivo. This testing may serve as a proof-of-concept that the tracer can detect osteomyelitis and may further provide data that leads to testing of the tracer in healthy volunteers and patients. Animal experiments are required to bridge the gap between laboratory inventions and the clinic [5].

Much can go wrong in the animal testing step, so it is wise to thoroughly consider the choice of model and the planning of experiments in advance. Some of the considerations before selecting a relevant animal model could relate to the aspects investigated, e.g., the quantitative or relative uptake of the tracer. The in vivo environment may result in additional tracer metabolites that were not present in in vitro experiments. Metabolites can be species-dependent, so it is important to consider, how close to human physiology is the physiology of the animal one plans to use, not only on a general basis but more specifically concerning the processes to be investigated (concerning infection and osteomyelitis). Does a potential uptake of radioactive metabolites affect the interpretation of the scan? How will the experiment results be interpreted? An incorrect choice of animal model or a misunderstanding of the effects can result in the discarding of a tracer that would have worked well in a human setting. One can also spend a lot of time and capital on a tracer that looks promising in preclinical studies but does not work well in humans.

To obtain the optimal information from preclinical work, experts are needed with many different skills. The process of manufacturing a new tracer is long. First, the tracer synthesis is optimized, and the quality control (QC) has to work. Second, if possible, in vitro experiments with the new tracer are performed [6]. Third, the tracer is tested in an animal model. Fourth, the tracer production is optimized to fulfil current Good Manufacturing Production (cGMP) [6]. Fifth, the tracer is applied to humans in a clinical setup [7]. Steps one to five are not a one-way street; it may be necessary to take a step back depending on the results obtained.

This review rests mainly on our own experiences as an interdisciplinary group of radiochemists, medical physicists, medical doctors, and veterinarians from 2013 to 2018. We worked intensively with preclinical tests of known and new osteomyelitis tracers in pigs (Figure 1). In the pig project, we performed 122 scans with 15 different radiotracers in 27 pigs to compare the performance of the various tracers. During our experiments, we obtained new knowledge on how to use our model. This resulted in a better and more robust model [8,9,10]. In the present review, we will generalize our experience and put it into a broader context. Where relevant, we have supplemented the review with a literature search (keywords used in PubMed in different combinations: animal models, osteomyelitis, tracer development, PET, imaging). However, it is important here to state that this is not a systematic review but is mainly based on our own practical experience as researchers, put in the context of a broader literature survey.

## 2. Choice of Animal Species

There is no single ideal animal species to use in osteomyelitis research, and every species has its advantages and disadvantages. Regardless of the importance of the animal species chosen, it is rare to find scientific papers which discuss, explain, and argue for the choice of the animal species used. However, promising preclinical results obtained in mice and other species often fail in clinical trials [11]. Physical size often matters when it comes to experimental animals, which are usually divided into two groups: small animals (mice and rats) and larger animals (rabbits, pigs, sheep, and dogs). This division is relevant in many aspects.

Small experimental animals such as rodents are typically inexpensive and easy to handle, and extensive knowledge is already available about their biology [12]. Inbred mice strains which are very uniform are available, which reduces the biological variation seen in animal (as in human) experiments. Thus, the calculated required group size is reduced. Transgenic mice with specific traits are also available. A disadvantage of using small animals is that they typically cannot be anesthetized for long periods and that their body size and blood volume severely limit their use for dynamic scanning studies [12]. For these reasons, it is challenging to compare different tracers in the same animal. This is compensated by using a higher number of animals, so the tracers are tested in various animal groups. That, however, conflicts with the reduction aspect of 3R [4]. Besides, the relatively poor spatial resolution of PET and SPECT scanners compared to the size of a mouse, or a rat, makes it hard to recognize details in the images [13]. Special microPET scanners for rodents have a slightly higher resolution. However, the better resolution of microPET scanners does not fully compensate for the inferior size of mice and rats [14]. It is also striking that bacterial inoculation usually has to be supplemented with sclerotic agents to achieve osteomyelitis in mice, rats, and rabbits.

The larger experimental animals, also called non-rodents, are often more expensive and difficult to handle [15]. Less knowledge exists about their biology, and it is almost impossible to obtain access to inbred strains, causing considerable variation in the results or the need for larger group sizes, which will also compromise the reduction aspect of 3R [15]. However, these larger animals enable detailed experimental work to be conducted and human clinical PET scanners to be used. The body size and blood volume of larger animals allow for dynamic studies, and they can often be kept anesthetized for a more prolonged period, which permits experiments comparing more tracers in the same individual [16]. Applying multiple tracers in the same animal will allow for a reduction in the group size. A general approach to evaluating animal models of, in our case, e.g., osteomyelitis is to judge their validity [17]. In the context of animal models for human physiology (here the pathophysiology of osteomyelitis), three forms of “validity” can be described [17]:

*Face validity* is defined by how well the animal model replicates the osteomyelitis phenotype in humans. 

*Construct validity* is how well the method used to induce osteomyelitis reflects the disease pathogenesis of human osteomyelitis. 

*Predictive validity* shows how well the animal model predicts new aspects of osteomyelitis in humans.

These and more aspects will be discussed in the following sections for the most commonly used animal species in osteomyelitis research: mice, rats, rabbits, pigs, sheep, and dogs. Others have reviewed the disease aspects of osteomyelitis in animals [18,19,20], but we will focus on the features that are important for testing tracers.

### 2.1. Mice

The small size of the mouse complicates its use for PET scanning due to difficulties in handling the injecting of the radioactive tracer and the resolution of the PET scan [14]. Nevertheless, several murine osteomyelitis models have been used to study, for example, the pathogenesis of osteomyelitis, the treatment of osteomyelitis with antibiotics, and hyperbaric oxygen therapy. In such studies, the phenotype has varied from that in humans, suggesting a lower face validity. Instead of PET scans, bioluminescent imaging may be performed. This technique takes advantage of the mouse’s small size which allows photons to pass through the tissue [21,22]. A simple tail vein inoculation model has been established where mice developed chronic osteomyelitis and could survive for at least 56 days [23]. Presumably, the mouse’s popularity is due to its accessibility and the considerable available knowledge of the mouse immune system, generally showing a good construct validity of the mouse as a model of human diseases. It is also an advantage that only small amounts of test substances are needed and that it is practically possible to prepare many mice with osteomyelitis in a short time. As the bodyweight is so low in mice, they present only limited lameness with limb osteomyelitis. That improves animal welfare, but from the point of face validity, it weakens the murine models.

### 2.2. Rats

The clear advantage of the rat compared to the mouse is the rat’s larger size. The size better compensates for the resolution of a microPET scanner. It is also an advantage that more blood samples can be collected, and anaesthesia can be extended [12]. The rat is, in general, a more robust animal (fewer deaths occur after inoculation) than the mouse, and also can maybe be anaesthetized for a longer time. Several osteomyelitis models have been developed using rats, and they are typically used to investigate the PET tracer uptake in osteomyelitis hot spots [24,25] or for prophylaxis and the treatment effects of antibiotics and other drugs [26]. The rat physiology is comparable to that of the mouse, but the knowledge of mice is more comprehensive [12].

### 2.3. Rabbits

The rabbit is the classic and oldest known model for osteomyelitis. Back in the 19th century, Rodet injected *Staphylococcus aureus* intravenously into a rabbit resulting in abscesses in the bones and many organs followed by death after a few days [27]. However, more than half a century passed before Scheman and co-workers succeeded in establishing a reproducible osteomyelitis model in rabbits by injection with bacteria and a sclerosing agent [28]. The added agent caused sclerosing of the vessels and tissue necrosis in the bones. In chronic osteomyelitis, to test the diagnostic value of [^18^F]FDG PET scans, the first model used was also a rabbit model [29]. Later, the use of [^18^F]FDG PET was further evaluated in an experimental osteomyelitis rabbit model after inoculation with *Candida albicans* [30]. In a long-term study, a group of researchers investigated the use of [^18^F]FDG for repeated diagnoses of chronic osteomyelitis of the tibia after inoculation with *Staphylococcus aureus* [31]. These [^18^F]FDG studies have shown a high predictive validity of the rabbit models. In addition to the rabbit’s historical significance, it is still widely used as an intermediate-sized animal. The rabbit is relatively easy to handle: its ear vein is easy to cannulate, it is easy to anesthetize for several hours, and its bone size makes it practically possible to induce osteomyelitis [32].

### 2.4. Pigs

Farm pigs sometimes spontaneously develop osteomyelitis, and the pathology seems to be very close to the osteomyelitis seen in humans, especially children [33,34], giving pig models a high face validity. Furthermore, the porcine immune system resembles that of humans in most aspects [35]. Osteomyelitis in pigs, as in humans, often derives from a bacterial infection in the blood. Besides a genetic and anatomical aspect, pigs have several similarities to humans [36]. Thus, pig models have a high construct validity. Pigs can be anesthetized for many hours [10], which combined with the size of pigs, allows for serial blood sampling [15]. This makes pigs attractive models for kinetic imaging studies.

The first pig model of osteomyelitis was established by inoculating bacteria in an ear vein, and microscopic osteomyelitis evolved as early as 12 h after [37]. However, the pigs in that study also developed pneumonia, as the lungs cleanse the blood for bacteria in pigs. In a large study in Denmark, we used pigs to compare established and new tracers for the detection of osteomyelitis [8,9,38,39]. These studies substantiated that the pig model has a high predictive validity, as the routinely used tracers in the human clinics (e.g., [^18^F]FDG) gave similar results in pigs. Our studies used a range of tracers in each animal. However, even though pigs can be kept anesthetized for many hours, intense blood sampling (>20 mL/kg) also affected the pigs’ physiology, with an effect on circulation and lung function [10].

### 2.5. Sheep

Sheep (and goats) are widespread models in orthopaedic research, and therefore they also have been used as osteomyelitis models. Kaarsemaker and co-workers established a suitable chronic osteomyelitis model by inoculating *Staphylococcus aureus* and a sclerosing agent into the proximal tibia marrow cavity [40]. This study showed a good face validity of the sheep model. The main problem with sheep, however, is that they are more difficult to anesthetize. There must be a change and reduction in feeding before anaesthesia to avoid gas accumulation in the rumen, which otherwise can cause life-threatening pressure on the lungs [41].

### 2.6. Dogs

While pigs and sheep are farm animals, dogs are pets. The corresponding emotional connotations make it problematic for many people (researchers and laypersons) to use dogs for research. Correspondingly, there are only a few studies on osteomyelitis induced in dogs. In one study, ten percent of the dogs died within two days after inoculation with *Staphylococcus aureus* together with a sclerosing agent in the hind limb artery [42]. Another study showed that it was possible to establish a model where half of the dogs survived for two years [43], which is promising for the face and construct validities in this model. There are no publications that have discussed using dogs specifically for tracer development in osteomyelitis. We therefore do not know the predictive validity of dogs. An advantage of the dog is that it is easy to handle and cannulate, and can be anaesthetized for many hours, while the disadvantage is that it is expensive and has a low genetic diversity. Labradors and beagles are the breeds most widely used as laboratory dogs. In middle-sized to large-sized dogs, there is a sufficient amount of blood for repeated measurements.

## 3. Induction of Osteomyelitis

Just as important as the choice of animal species is the choice of method of inducing osteomyelitis. As mentioned earlier, construct validity describes how well the induced osteomyelitis reflects the pathogenesis of osteomyelitis in humans. Bacteria that are spread haematogenously are a common cause of osteomyelitis, especially in children [44,45], while prosthesis-related osteomyelitis is more common in adults [46]. In humans, it is frequently the tibia and femur that most often become infected. The most common causative bacterium is *Staphylococcus aureus* [44].

Accordingly, most animal models are based on the induction of osteomyelitis in the tibia with *Staphylococcus aureus*. Osteomyelitis can be induced in the laboratory by local inoculation of the bacteria or through haematogenous injections into blood vessels. One of the oldest methods is the tibial model, first performed in rabbits, where the bacteria are injected into the tibial medullar cavity by using a needle [47]. Osteomyelitis can also be applied to other bones, such as in femoral, radial, mandible, or spine models. A local implantation of bacteria-contaminated prostheses has also been used [48]. The direct injection of bacteria into a bone has the advantage that the location of osteomyelitis is known, and the risk of spread is limited, although not controlled. However, the method is traumatic. The bacteria can also be injected into a vein and thereby, in addition, cause the experimental animal to develop other infections, including septicaemia [27]. A more sophisticated method is to inject the bacteria into a local artery, whereby the bacteria are trapped in the capillaries so that local osteomyelitis develops. Osteomyelitis can occur in all the bones that the selected artery supplies with blood. In addition to osteomyelitis, soft tissue infections will also emerge to a certain extent [49]. This method is also not perfect, as some bacteria will pass through the capillaries and thereby be able to settle in other organs. For example, we showed that in young, domesticated pigs, septicaemia developed more frequently when pigs weighing 40 kg were used than when pigs that were a few weeks younger, weighing 20 kg, were used [9]. Another disadvantage of arterial inoculation is that it requires a surgical procedure rather than a general intravenous inoculation, which can be performed non-invasively in superficial veins of awake or sedated animals [9]. Still, the arterial-inoculation method potentially provides better animal welfare than the vein-inoculation method by reducing the risk of septicaemia. Furthermore, whole-body PET scans of arterial inoculated laboratory animals can give anatomical information about the location in the body of the bacterial infection.

The origin of the bacteria used also differs between studies. Two different perceptions exist. Some researchers prefer using strains isolated from human osteomyelitis, as they are the most relevant. Others, however, prefer animal strains, which are adapted to cause infections in the animal species used. A review of animal models of *Staphylococcus aureus* osteomyelitis reported that several different human and animal isolates have been used and reported in published papers [18]. This causes difficulties in comparing studies. In line with this, the inoculation doses (CFU/kg) are also very different from study to study. The chosen bacteria strain will probably affect the results of the tracer testing, as it has unique virulence factors. Higher inoculums of bacteria will shorten the time to osteomyelitis onset, but it will also aggravate the clinical symptoms with poor animal welfare as a result. In general, the inoculation doses (CFU/kg) needed are higher in small than in larger animals and higher for human bacteria strains than for species–specific bacteria strains [50]. Moreover, the age of the animals should be considered. As osteomyelitis most often occurs in children, young animals are most often used as models. However, it appears that adult individuals were used in the dog studies [42,43].

## 4. Analgesia and Humane Endpoints after Inoculation

The laboratory veterinarian should always be consulted before initiating animal experiments. Below we present some of the general aspects of analgesia and humane endpoints to be considered. It must always be recognized that osteomyelitis and other infection models can have a severe impact on animal welfare (the refinement aspect of the 3R) [51]. Humane endpoints for the euthanasia of the animals must be properly defined before initiating the experiments. This will generally also be required to obtain authorization for the experiment. For both the animals and the experiments, it is preferable if the endpoints are never reached. The timely application of analgesia will often be a great help in this respect. In our experience, the clinical signs of osteomyelitis can emerge within a few hours in pigs [8]. Therefore, the animals should have analgesics just after the induction of osteomyelitis, and it is best to continue the treatment until the time of euthanasia at the end of the experiment. In addition to securing animal welfare, this will also ensure a uniform treatment of the experimental animals. The choice of analgesic should be based on its ability to induce sufficient analgesia, but it should also be considered how the drug affects the development and diagnosis of osteomyelitis. That should, therefore, always be discussed with the laboratory animal veterinarian. The veterinarian has to comprehend both the analgesic properties of the drugs and the possible effects on the research results. Non-steroidal anti-inflammatory drugs (NSAIDs) inhibit the immune system by inhibiting the COX enzymes and may also affect the current infection. However, NSAIDs are widely used as the immunosuppressive effect is weak and because the administration can be reduced to 1–2 times per day, which facilitates their practical use [52]. In selected cases, short-term treatment with antibiotics should be considered to balance the NSAID treatment [8] or to prevent septicaemia [40]. Care should be taken as both antibiotics and NSAIDs may affect the results of the investigations. Opioids can be used as an alternative as they act on the pain perception in the brain, and some studies have shown some immunomodulatory effects from the opioid treatment used in experimental animals [53]. It is a disadvantage that opioids have a shorter half-life than NSAIDs [54]. In theory, local analgesia could also be used, as this combines a good analgesic effect with little risk of interaction with the research results. In practice, however, it is challenging to apply local analgesia repetitively to experimental animals and the animals need continuous monitoring. For example, body temperature should be monitored and controlled, as anesthetized experimental animals may quickly become hypothermic, or they may be hyperthermic due to the inflammatory condition. The humane endpoints defined before initiating the experiment determine when treatment should be initiated and when the animal should be euthanized. The variable data expressing humane endpoints can be advantageously managed in a scoring form used for the continuous monitoring of animal conditions. Either standard scoring scheme (e.g., well-being, growth rate, activity level) can be used for the animal species [55]. Specific scoring schemes can also be established for osteomyelitis, arranged according to animal species and the induction method of osteomyelitis (e.g., degree of lameness, body temperature, respiration rate, feed intake). In the case of hematogenous induction, there is a risk of bacteria spreading to other organs, inducing septicaemia, and, therefore, the humane endpoints should include monitoring for fever, cough, anorexia, and tachycardia, in addition to the local signs in the form of lameness, swollen joints, and movement.

## 5. Tracer Evaluation Based on the Scan Data

New osteomyelitis tracers are typically evaluated in anesthetized animals and should include animals with and without osteomyelitis. It can be advantageous to compare the scan of the evaluated tracer with a scan using a more well-established tracer (such as [^18^F]FDG). For new tracers, we are typically interested in the effectiveness of the new tracer in finding osteomyelitis lesions compared to the performance of well-established tracers [8]. For a new tracer to have a future in clinics, it must have other advantages over the established ones. 

Such a comparison is particularly convenient with larger animals, as they can typically be anesthetized for a long enough time to be scanned with more than just a single tracer. Mice and other smaller experimental animals can only be anesthetized for a shorter period. The inclusion of computed tomography (CT) in the scan sequence gives more exact information on the anatomical location and number of osteomyelitis lesions. The PET imaging acquisitions can be a static scan, performed after a certain period once the tracer has been injected [8]. However, to obtain more detailed information, PET imaging can be recorded dynamically. Dynamic scans give information on tracer kinetics [49]. In the latter case, a series of blood samples are drawn, and measurements of radioactivity content are performed to establish time–activity curves, which are needed for a kinetic analysis of the tracer behaviour in the body. A normal PET scanner has a field-of-view of 20 cm in which it can obtain a dynamic scan.

It is also possible to first record a dynamic scan focused on the part of the animal with infection and then finish with a whole-body static scan—the dynamic scans can be used to determine the correct time after the tracer injection to record static scans. The main drawback of a dynamic PET scan is that the scanner does not span the whole length of larger animals but only ca. 20 cm (depending on the scanner type). Blood samples are usually also required for analyses of metabolites of the tracer. Blood sampling is facilitated by arterial catheters through which the blood can be taken [56]. It is also possible to sum all the time frames from the dynamic scan into a single static image showing the time-averaged uptake over these time frames.

Before starting the scanning, it is critical to fix the body part that is being scanned to eliminate body movements during the scan, especially during prolonged scans, such as dynamic scans starting at tracer injection and ending an hour or more later. Fixation can be completed in several ways. For pigs, we have developed a holder, which is shown in Figure 2**.** The holder allows for the fixation of the hind limbs of the anesthetized pig, and it is possible to place the contralateral hind limb in an almost identical position as the limb with osteomyelitis so that the non-infected limb can function as a healthy control. In the published papers, fixation methods are rarely described.

For all promising tracers, whole-body scans are also performed and used for the calculation of dosimetry data. An estimation of dosimetry will generally be required to obtain approval for testing the tracer in the first human subjects and patients. After completing all the scans, the experimental animals are euthanized, and it is beneficial to perform a regular necropsy of the whole body or at least of the body part with osteomyelitis. It confirms the presence of osteomyelitis and allows re-isolation of the injected bacteria in the places indicated by the imaging. Necropsy is also used to clarify the findings in scans.

## 6. Discussion, Conclusions, and Perspectives

The literature on animal models of osteomyelitis extensively focuses on pathology, drug testing, and immune response. This review presents the topic from the perspective that models should be used for testing new PET tracers. A new PET tracer starts with a chemical synthesis in the laboratory, involving the production of the molecule and labelling it with a radioactive isotope. The discovery and development process of a new radioactive drug is not significantly different from other drugs. However, there is one significant difference: any pharmacological effects of a radioactive drug to be used for diagnosis are unwanted. One would like to examine and visualize a biological system without affecting it. It is often a little difficult, but fortunately, PET and SPECT scanning are incredibly sensitive in the sense that only small amounts of tracer need to be injected. That means that one can most often ignore the pharmacological effects of the injected substance. 

During the process of inventing and testing potential new drugs, it is important to understand the underlying biochemistry and biological system of the disease. In radioactive drug development, we often rely on and use the discoveries achieved in traditional, non-radioactive drug development. The investigation continues with in vitro experiments, e.g., involving cell cultures or tissue samples to find the initial binding affinity and the degradation profile of the tracer. The distribution coefficient (logD) is a parameter relatively simple to obtain in the laboratory, and it will offer information if the potential tracer is water or fat-soluble. If logD is in the correct range, the tracer may work. However, if it is far from optimal it indicates that the tracer most likely will not work. The focus should then be on changing the polarity of the potential tracer rather than proceeding with animal studies. If successful, tracer development advances in relation to in vivo testing in animals may finally reach the stage of human application. The tracer may be viewed as a baton handed from one runner to the next—from a medical doctor to a biochemist, to a chemist to a radiochemist, to a biologist to a veterinarian, and back to the medical doctor—allowing each member of the team to use their competence. All members of the team should work together and exchange all their knowledge and skills. Specifically, the chemist should not solely stick to the laboratory but should also be part of the remaining steps. QC, a proper handling of the tracer, an analysis of metabolites, and an evaluation of the results can involve chemists, physicists, veterinarians, and medical doctors. For such a team to work best, all members must have a minimum basic knowledge of the others’ fields. In this review, we have focused on animal testing, a relevant aspect of tracer testing which may be new to many chemists.

## Figures and Tables

**Figure 1 molecules-26-04221-f001:**
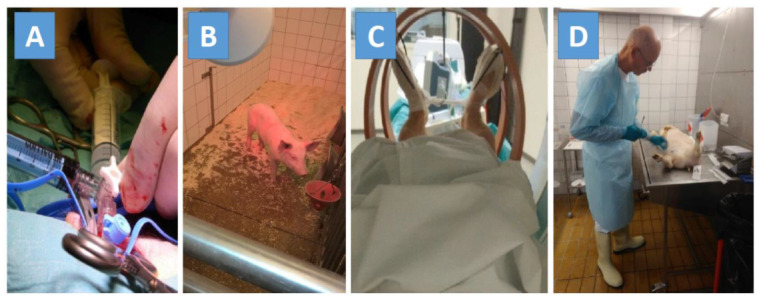
Step-by-step procedures for haematogenously induced osteomyelitis in a pig model used for comparing new tracers with known tracers: (**A**) Inoculation with a porcine isolate of *Staphylococcus aureus* in the right femoral artery of anesthetized 20 kg domestic pigs. (**B**) For the next seven days, the pigs were treated with painkillers and their clinical signs were scored every eight hours. If signs of bacteria spread to internal organs (cough, severe fever, tachycardia, anorexia, or inactivity) and thus septicaemia was observed, the pig was euthanized as a result of the pre-set humane endpoints. (**C**) After six days the pigs were CT scanned for verification of visible osteomyelitis, and the next day (day seven after inoculation) the pigs were exposed to the PET scanning program with several known and new osteomyelitis tracers. Both dynamic and static scans were performed, and blood samples from the carotid artery were collected and used to establish blood–activity curves and metabolite analyses. (**D**) At the end of the scans, the pigs were euthanized during the anaesthesia with an overdose of pentobarbital, after which a regular necropsy of the pigs was performed. This was guided by the findings from the CT scans the day before. The necropsy was performed to disclose the location and character of the various induced lesions so that the effectiveness of the tested tracers could be evaluated relative to the necropsy findings. Bacteria from lesions were cultured to determine if they originated from the inoculated bacteria strain.

**Figure 2 molecules-26-04221-f002:**
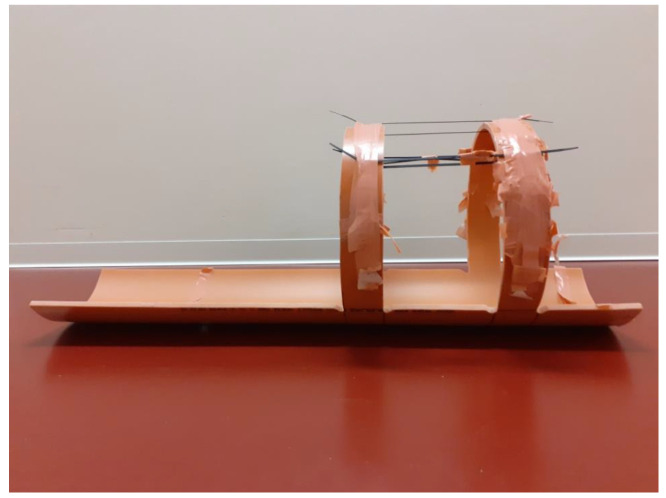
This homemade holder for the fixation of the pig’s hind limbs was made from a sawn-through PVC plastic tube. It is possible to fix both hind limbs with tape within the scanner’s field-of-view, which is the distance between the arches, so that the healthy hind limb can act as a control. The holder is also seen in use in Figure 1C.

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
