# Peer review of "Preclinical Testing of Radiopharmaceuticals for the Detection and Characterization of Osteomyelitis: Experiences from a Porcine Model"

_molecules, 2021, doi:10.3390/molecules26144221_

Round 1
Reviewer 1 Report
Alstrup and coworkers reported preclinical testing of new PET tracers for osteomyelitis. In this short-review, the authors intend to present the models which could be used for testing new PET tracers. However, they have provided less details on PET tracers and their roles in researches related to animal models of osteomyelitis which is a lack of point in this paper. Even, the discussion part seems poor. Few of the comments include:
- In lines between 228 – 237,( In subtitle: 2.6 Dogs), the authors have not described the contect of “dog” animal models for human physiology in terms of three forms of validity which they have already indicated in the manuscript. They should insert knowledge on it from the literature.
- In section “5. Evaluation of the tracers”, the authors have not provided enough knowledge on the types of tracers, their production methods, cytotoxicity features, and how their features affect scanning performance in PET.
- In lines between 373 – 376, the authors provided a detail on why fixing the body part is important for collecting a better scanning results and given only their own design to perform fixation procedure. They should also refer other types of the fixation apparatus what they are with a citation from the literature.
Author Response
Alstrup and coworkers reported preclinical testing of new PET tracers for osteomyelitis. In this short-review, the authors intend to present the models which could be used for testing new PET tracers. However, they have provided less details on PET tracers and their roles in researches related to animal models of osteomyelitis which is a lack of point in this paper. Even, the discussion part seems poor. Reply: We thank you for your comments. In the same issue of Molecules, we have already published a review on the existing radiotracers used for bone marrow imaging, and therefore our present manuscript does not address this. We have now made this clearer in the text, and inserted a reference to our printed paper for interested readers. Furthermore, we have revised the discussion (see the following comments), and hope it is satisfying. Few of the comments include: 1. In lines between 228 – 237,( In subtitle: 2.6 Dogs), the authors have not described the contect of “dog” animal models for human physiology in terms of three forms of validity which they have already indicated in the manuscript. They should insert knowledge on it from the literature. Reply: Thank you for this comment and we agree. As only little is known about this, we had only included a few extra pieces of information on face, construct, and predictive validity of dogs used for preclinical testing of new osteomyelitis tracers. Furthermore, we have added more information on relevant dog physiology.
2. In section “5. Evaluation of the tracers”, the authors have not provided enough knowledge on the types of tracers, their production methods, cytotoxicity features, and how their features affect scanning performance in PET. Reply: We thank the reviewer, but in line with the other reviewer’s suggestions, we have focused the manuscript on our experiences with research animals. A reference (our recent review in Molecules by Jødal L et al.) has been added, as this focuses on the osteomyelitis tracers. 3. In lines between 373 – 376, the authors provided a detail on why fixing the body part is important for collecting a better scanning results and given only their own design to perform fixation procedure. They should also refer other types of the fixation apparatus what they are with a citation from the literature. Reply: We agree, but based on the literature, this is not a point the authors have mentioned in their papers. We certainly believe that this has been a topic that the research groups have considered when carrying out the experiments, but they have not reported it in their papers. This has now been added to our manuscript. We have also added more information about our fixation method in the legends to Figure 2.
Reviewer 2 Report
This is a review about experimental models of preclinical testing of new PET Tracers for osteomyelitis. Below are my comments:
- You have stated in the introduction section that this review is partly based on literature and partly on your experience. In addition, you have various papers on this subject published before. In my opinion, it will be better to present this review from your point of view, as researchers with experience in this field, in comparison with methods used by others, with advantages and disadvantages. It is somehow difficult to understand your exact contribution (as actual researchers) in this paper.
- There is no flow chart showing selection of papers for the review and the criteria you have used.
- Figure 1. Part A – it should be better to show an exact photo of the inoculation process, the devices used for arterial puncture, instead of a picture of someone doing something in a sterile way.
- Maybe the title should be changed to better fit the content – you actually have performed a comparison of different model animals used for the development of radioactive tracers
- Line 35 – the word “animals” seems too much after saying “animal osteomyelitis model”
- Line 38 – every phrase begins with ```Osteomyelitis…` - please change
Author Response
This is a review about experimental models of preclinical testing of new PET Tracers for osteomyelitis. Below are my comments: 1. You have stated in the introduction section that this review is partly based on literature and partly on your experience. In addition, you have various papers on this subject published before. In my opinion, it will be better to present this review from your point of view, as researchers with experience in this field, in comparison with methods used by others, with advantages and disadvantages. It is somehow difficult to understand your exact contribution (as actual researchers) in this paper. Reply: Thank you for these very constructive comments. We have discussed it, and we agree that the focus should be more about our own experience as researchers working with preclinical testing of osteomyelitis tracers. Therefore, the title of the manuscript has been changed, so it is clear that the focus is on our own experience with preclinical testing of osteomyelitis tracers, and we also make it clear throughout the manuscript. We believe it is now more clear what our contributions are. 2. There is no flow chart showing selection of papers for the review and the criteria you have used. Reply: According to the previous comment from reviewer 1, we have made changes in order to clearly show that the manuscript is based on our own experiences, and therefore only secondarily on the literature, and so we have not included a flow chart. However, we have introduced some lines
which describe the criteria for the choice of literature. This should replace a flow chart. 3. Figure 1. Part A – it should be better to show an exact photo of the inoculation process, the devices used for arterial puncture, instead of a picture of someone doing something in a sterile way. Reply: We agree that a close-up of the inoculation is a better choice. This has been added. Therefore, Figure 1, Part A has now been changed. 4. Maybe the title should be changed to better fit the content – you actually have performed a comparison of different model animals used for the development of radioactive tracers Reply: We agree and have now rephrased the title: “Preclinical Testing of Radiopharmaceuticals for Detection and Characterization of Osteomyelitis: Experience from a Porcine Model”. 5. Line 35 – the word “animals” seems too much after saying “animal osteomyelitis model” Reply: We agree and have now deleted “animals”. 6. Line 38 – every phrase begins with ```Osteomyelitis…` - please change
Reply: We agree and have now replaced the second “Osteomyelitis” with “It”.
Round 2
Reviewer 1 Report
I am satisfied with the corrections undertaken by the authors of this mini-review paper entitled: "Preclinical Testing of new PET Tracers for Osteomyelitis" (even with its new title in the revised version). As such the study can now be published in this prestigious journal.